# The ORIGIN Space Instrument for Detecting Biosignatures and Habitability Indicators on a Venus Life Finder Mission

Niels F. W. Ligterink [1],*[iD], Kristina A. Kipfer [1][iD], Salome Gruchola [1][iD], Nikita J. Boeren [1,2][iD], Peter Keresztes Schmidt [1][iD], Coenraad P. de Koning [1][iD], Marek Tulej [1], Peter Wurz [1,2][iD] and Andreas Riedo [1,2],*[iD]

1 Physics Institute, Space Research and Planetary Sciences, University of Bern, Sidlerstrasse 5, CH-3012 Bern, Switzerland; kristina.kipfer@unibe.ch (K.A.K.); salome.gruchola@unibe.ch (S.G.); nikita.boeren@unibe.ch (N.J.B.); peter.keresztes@unibe.ch (P.K.S.); coenraad.dekoning@unibe.ch (C.P.d.K.); marek.tulej@unibe.ch (M.T.); peter.wurz@unibe.ch (P.W.)
2 NCCR PlanetS, University of Bern, Gesellschaftsstrasse 6, CH-3012 Bern, Switzerland
* Correspondence: niels.ligterink@unibe.ch (N.F.W.L.); andreas.riedo@unibe.ch (A.R.)

**Abstract:** Recent and past observations of chemical and physical peculiarities in the atmosphere of Venus have renewed speculations about the existence of life in its clouds. To find signs of Venusian life, a dedicated astrobiological space exploration mission is required, and for this reason the Venus Life Finder mission is currently being prepared. A Venus Life Finder mission will require dedicated and specialized instruments to hunt for biosignatures and habitability indicators. In this contribution, we present the ORIGIN space instrument, a laser desorption/laser ablation ionization mass spectrometer. This instrument is designed to detect large, non-volatile molecules, specifically biomolecules such as amino acids and lipids. At the same time, it can also be used in ablation mode for elemental composition analysis. Recent studies with this space prototype instrument of amino acids, polycyclic aromatic hydrocarbons, lipids, salts, metals, sulphur isotopes, and microbial elemental composition are discussed in the context of studies of biosignatures and habitability indicators in Venus's atmosphere. The implementation of the ORIGIN instrument into a Venus Life Finder mission is discussed, emphasizing the low weight and low power consumption of the instrument. An instrument design and sample handling system are presented that make optimal use of the capabilities of this instrument. ORIGIN is a highly versatile instrument with proven capabilities to investigate and potentially resolve many of the outstanding questions about the atmosphere of Venus and the presence of life in its clouds.

**Keywords:** space instrumentation; Venus; atmosphere; astrobiology; origin of life; prebiotic chemistry

## 1. Introduction

Venus is a hellish planet. Temperatures on its surface readily reach 730 K [1] and, historically, it may never have been cold enough to condense water on its surface to form oceans [2]. Its dense atmosphere, with a pressure of 9.3 MPa at the surface, almost 100 times what is experienced on Earth, is dominated by carbon dioxide ($CO_2$) with minor components of nitrogen ($N_2$), sulphur dioxide ($SO_2$), COS species, and hydrogen chloride (HCl) (e.g., [3–5]), while the temperature is lower in the cloud layers, even reaching temperate levels in the lower and middle cloud layers between 47.5 and ~60 km altitude [6], the conditions are not necessarily more hospitable. Venus's atmosphere is drier than the most arid locations on Earth. The small amounts of water vapour present combine with large amounts of sulphuric acid ($H_2SO_4$) to form extremely acidic droplets [7]. At first sight, life has a small chance of emerging and surviving in this environment.

Nonetheless, the presence of life in Venus's clouds has been proposed (e.g., [6,8,9]). A Venusian aerial biosphere could explain a number of peculiar observations of this planet. High molecular oxygen ($O_2$) concentrations were found in Venus's cloud deck [10]. At present, no abiotic processes are known to produce $O_2$ locally and in these concentrations,

but it could be a byproduct of an ammonia ($NH_3$) formation scheme [11], possibly by organisms. Furthermore, the clouds of Venus strongly absorb UV light in the 320–500 nm range by an unknown absorbing species, which could, among others, be iron chloride ($FeCl_3$) dissolved in sulphuric acid droplets (see for a review [12]). However, biomolecules, such as nucleic acids, iron-containing proteins, or photosynthetic pigments also absorb in this wavelength regime and could be responsible for the observed UV absorbance, or contribute to it [13]. Finally, the detection of phosphine ($PH_3$) in Venus's atmosphere was recently reported [14]. Although this observation has been challenged (e.g., [15]), the potential abundance of $PH_3$ cannot currently be explained by any known chemical processes, necessitating the invocation of a previously unknown mechanism, which could include biological activity [16]. The above reasons not only make Venus a fascinating Earth-twin with a run-away greenhouse effect, but perhaps also a promising astrobiological target to conduct the search for extraterrestrial life.

Continued studies of Venus are needed to understand its possible habitability and to search for signs of life, especially with up-close space exploration missions. Previous space missions have obtained some of the most detailed and sensitive measurements of Venus, for example with Venus Express (ESA [17]) and Akatsuki (JAXA [18]), and even studied the planet in situ with probes during the Venera and Vega (Soviet Space Program) and Pioneer (NASA) programs. In the next decade, a number of missions will fly to Venus, including DAVINCI+ and VERITAS (NASA [19,20]) and EnVision (ESA [21]). While these missions will contribute to our understanding of Venus's history, its surface, and its atmosphere (e.g., [5,22]) and may shine light on Venus's habitability and possible presence of life, they are not astrobiologically focused missions. The Venus Life Finder (VLF, MIT/Breakthrough initiatives [23]) mission stands in stark contrast to the previous missions. This mission is aimed at taking a deep dive into Venus's atmosphere with the specific aim of assessing the habitability and searching for signs of life in its temperate cloud layer. Various VLF mission concepts of different sizes and capabilities are proposed, such as a small descent probe, a larger balloon-born mission, or a sample return mission.

A space exploration mission like VLF requires instruments that are up to the task of detecting habitability indicators and biosignatures in the clouds of Venus. Fluorescence microscopy, for example, can be used to identify microbial cell structures, while a mass spectrometer can be used to identify biosignature gases such as $PH_3$ and $NH_3$ [23]. However, many molecular, atomic, and isotope biosignatures are too small to be visualized with microscopes or not volatile enough to enter the gas-phase where they can be detected with a regular neutral gas mass spectrometer or even less by a GC-MS. This will mean that many important biomolecules, such as amino acids, nucleobases, and lipids, and also other biosignatures or habitability indicators, such as salts, metals, sulphur isotope fractionation, and polycyclic aromatic hydrocarbons (PAHs), will not be analysed. Therefore, the VLF Venus Airborne Investigation of Habitability and Life (VAIHL) mission concept, a large and highly capable balloon-born mission, will be outfitted with a large instrument suite, including a laser-based mass spectrometer (LMS). Laser-based mass spectrometry is a sensitive and versatile series of measurement techniques that is particularly suitable for space exploration missions [24,25]. In this contribution, we present the ORIGIN space instrument [26], an LMS considered for use on a VLF mission. We note that throughout this manuscript we will present ORIGIN for a VLF mission but specifically mean the VAIHL mission concept.

ORIGIN (ORganics Information Gathering INstrument, [26]) is under development at the University of Bern and is intended to be used on in situ space exploration missions. It can operate in laser desorption/ionization mass spectrometry (LDMS) and in laser ablation/ionization mass spectrometry (LIMS) mode. In its default LDMS mode, it is designed to search for biomolecules, such as amino acids and lipids. By operating in LIMS mode, it can also analyse elemental compositions and isotope ratios. In this contribution, we describe ORIGIN and how it can be used for VLF science. An overview of the instrument is given in Section 2. Section 3 presents various biosignatures and habitability indicators, such

as biomolecules, aromatic hydrocarbons, metals, single microbe detection, and sulphur isotope fractionation, that have been measured with ORIGIN in various performance studies and will be useful to search for life in the atmosphere of Venus. New ORIGIN measurements are presented of salts that are relevant to Venus's cloud chemistry, such as ammonium sulphate and calcium carbonate. Advanced analysis techniques used on ORIGIN data are presented in Section 4. Implementation of ORIGIN into a VLF mission and search methodologies in the clouds of Venus are discussed in Section 5. Finally, conclusions are given in Section 6.

## 2. ORIGIN Overview

The ORganics Information Gathering INstrument system (ORIGIN) is a prototype LMS instrument, designed for in situ measurements on planetary surfaces or in planetary atmospheres. The instrument has been described in detail in Ligterink et al. [26]. The system is optimized to operate in LDMS mode and detect non-volatile molecular compounds. This can be done by performing a direct measurement of a solid sample or on the residues that are left when the solvent of a liquid extract is evaporated. In the context of Venus's clouds, the sample could be collected sulphuric acid droplets, from which the $H_2SO_4$, $H_2O$, and other volatile species are evaporated.

The laboratory prototype consists of three core components: a laser desorption/ionization source, a mass analyser, and a sample handling system. These core components will also be present in an ORIGIN flight model. A schematic description of an ORIGIN flight model is presented in Section 5.

To desorb and ionize sample material, ORIGIN uses a pulsed Q-switched Nd: YAG laser ($\lambda$ = 266 nm, pulse duration $\sim$ 3 ns, pulse repetition rate = 20 Hz). Laser pulses pass through various optical components to measure the pulse energy and subsequently guide it through a beam expander and a lens system to focus the laser pulses on the sample holder (focal spot size $\varnothing \sim$30 μm). The laser can be operated at different laser pulse energies, which generally range from 1 to 4 μJ. At much higher pulse energies of around 10 μJ, the laser irradiance becomes high enough to ablate material, and therefore, ORIGIN can also operate in a laser ablation/ionization mode.

The laser beam is guided through a quartz window installed on top of a vacuum chamber, which houses the mass analyser, and is maintained at a pressure of $< 5 \times 10^{-8}$ mbar. A lightweight and miniature (160 mm $\times$ $\varnothing$ 60 mm) reflectron-type time-of-flight mass spectrometer is housed in the vacuum chamber. This mass analyser is operated in positive ion mode, meaning it currently only detects cations. However, there is no technical limitation to operating it in negative ion mode and detecting anions. In desorption mode, the mass analyser achieves a mass resolution of m/$\Delta$m = $\sim$1'000 and a dynamic range of up to eight orders of magnitude. A high-voltage power supply sets the voltages on the ion optics and micro channel plate detector stack, while a high-speed analog-to-digital converter card records the mass signals.

Laser pulses traverse co-axially through the mass analyser, exit at the bottom of the mass analyser, and are focused slightly above the sample holder. The sample holder is placed on an XYZ-stage. In the Z-direction, the sample can be moved in and out of the laser focus, while in the XY-plane it can be moved to different samples or different locations of the same sample can be analysed. The sample holder itself is typically made of stainless steel with various cavities (usually $\varnothing$ 3 mm) into which liquids can be drop cast. However, other sample holder materials such as gold-coated glass wafers have also been used.

ORIGIN can analyse the chemical composition of a sample with a single laser pulse, on a single position on the sample. However, it is more useful to operate it in so-called grid mode. In this mode, the sample material is then moved in the XY-plane to analyse the chemical composition of the sample on multiple positions, usually with multiple laser shots per position. A typical ORIGIN measurement involves a linear grid of forty analysed positions at $\sim$50 μm step size, while each position is probed with 100 laser pulses, resulting in 4'000 acquired mass spectra. Different grid patterns can be set, only limited by the

mechanics of the stage. A secondary limitation is the generated data volume, which can be substantial if all individual spectra were to be transmitted and much more than the downlink data volume of the VLF can handle (a maximum downlink of 500 MBits per VLF circumnavigation of Venus is given in [23]). It is therefore essential to decrease the data volume, and this can be performed by online data processing such as histogramming to, for example, reduce 4'000 spectra to a single one. Data compression routines can be employed to further reduce the data volume for transfer back to Earth.

The reason for opting for grid scanning has to do with the nature of the sample material. Solid residues are obtained by evaporating the solvent of a solution. A residue formed in this way is not homogeneous and contains concentration gradients. Analysing this sample material with a single laser pulse thus poses the problem that some locations might be devoid of material, seemingly resulting in a non-detection. Just as important is quantification by correlating the recorded LDMS signals with the average surface concentration. A single laser pulse scheme runs the risk of analysing a high, above average concentration patch of material, thus resulting in a misleading reading of the concentration. In mixtures, certain molecules might even segregate, thus exacerbating the problem of single position analysis. Grid scanning reduces all these problems by sampling a larger area of the sample and summing its mass spectrometric signals. In this way, all molecules that are present above the limit of detection, but which are spread out on different locations of the sample surface during sample preparation, can be detected. At the same time, it gives a robust analysis of the concentration of compounds in a sample by determining the average surface concentration. This is essential for determining the abundances of molecules and to identify unusual variations that may indicate biotic production mechanisms.

## 3. ORIGIN Analysis of Biosignatures and Habitability Indicators

ORIGIN excels in the analysis of non-volatile and refractory molecules, such as amino acids, lipids, polycyclic aromatic hydrocarbons (PAHs), and salts in its LDMS desorption mode. At higher laser pulse energies, it can also be used for elemental and isotope analysis of solids in LIMS ablation mode. While its mass analyser can be repurposed to analyse environmental gases such as $NH_3$ and $PH_3$, this comes at a significant performance cost due to the addition of an extra ionization method and the modification of ion optics, and is thus not recommended. In the following sections, ORIGIN's performance in detecting each of these signatures is discussed in the context of VLF science.

### 3.1. Biomolecules

In the ensemble of biomolecules, ORIGIN performance analysis efforts have thus far focused on amino acids and lipids, which are essential building blocks for life as we know it. Ligterink et al. [26] demonstrated that ORIGIN could identify 14 proteinogenic (Glycine, L-Alanine, L-Serine, L-Valine, L-Threonine, L-Leucine, L-isoLeucine, L-Aspartic acid, L-Glutamine, L-Methionine, L-Histidine, L-Phenylalanine, L-Tyrosine, and L-Tryptophan) and 4 abiotic amino acids ($\gamma$-Aminobutyric acid, L/R-$\alpha$-Aminobutyric acid, $\alpha$-aminoisobutyric acid, and L-$\beta$-Aminobutyric acid.) based on their simple fragmentation patterns in LDMS. Several of these ORIGIN mass spectra are shown in Figure 1. The fragmentation patterns were discovered to be dominated by masses caused by the loss of a carboxyl (-COOH) or a side chain group, such as $m/z = 30$ ($NH_2CH_2^+$) of glycine (75 u) or $m/z = 107$ ($HO\text{-}C_6H_4\text{-}CH_2^+$) of tyrosine (181 u). The quantification of amino acids at average surface concentrations between 0.1 and 10 pmol mm$^{-2}$ was shown to be possible. Three sigma limits of detection (LOD$_{3\sigma}$) ranged from 700 fmol mm$^{-2}$ for glycine to about 1 fmol mm$^{-2}$ for tryptophan. ORIGIN has a high sensitivity to amino acids with aromatic ring structures, such as tryptophan, tyrosine, and histidine. Aromatic structures generally absorb well in the deep UV and, therefore, the 266 nm laser pulses of ORIGIN readily interact with these molecules. Furthermore, aromatic rings remain comparatively stable when ionized, ensuring longer longevity before dissociation and hence better detectability.

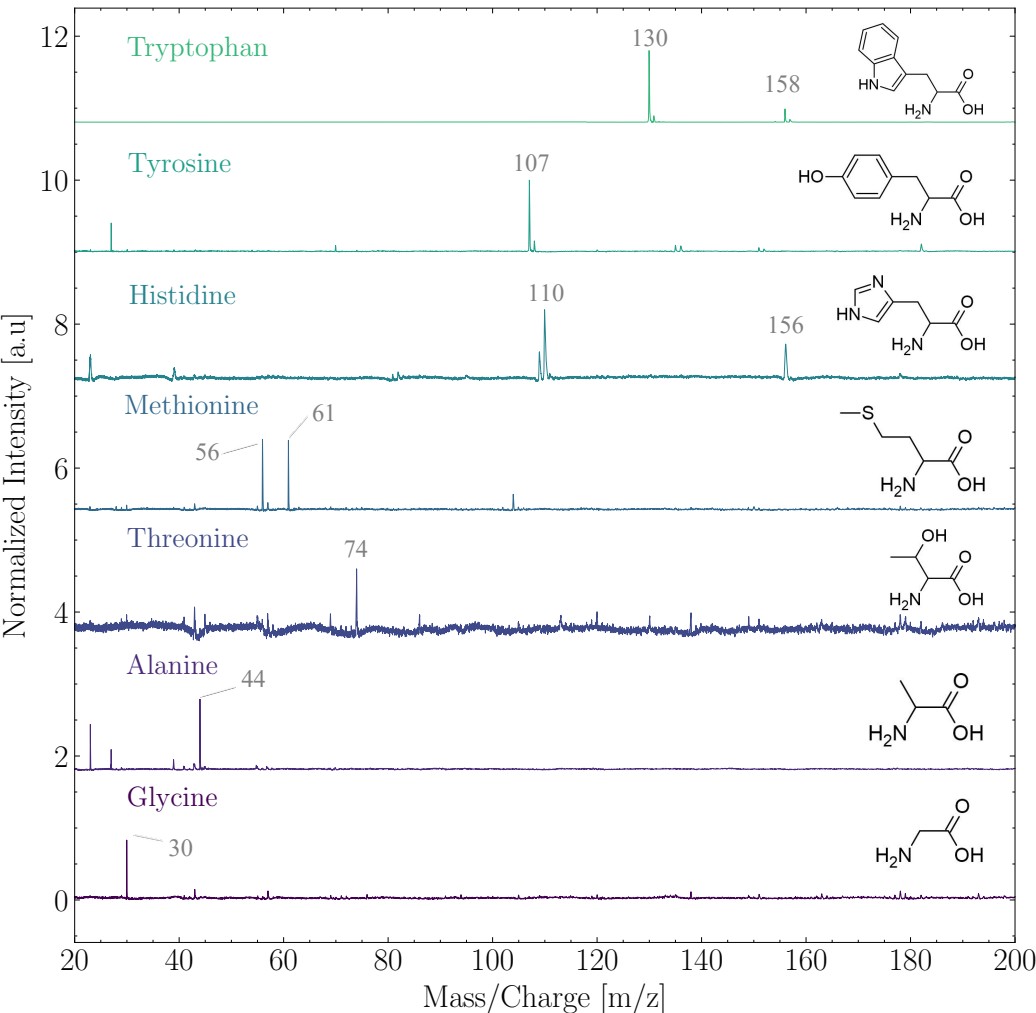

**Figure 1.** Normalized spectra of the amino acids tryptophan, tyrosine, histidine, threonine, alanine, and glycine. The structure of the respective amino acid has been added to the right and prominent mass peaks are indicated. The spectra have been offset from each other. All measurements were conducted at pulse energies between 2 and 3 μJ at an average surface coverage of 14.1 pmol mm$^{-2}$.

Multiple lipids were studied to further demonstrate the measurement capabilities of ORIGIN (Boeren et al. in prep.). Mass spectra of two different lipids, phylloquinone (vitamin K$_1$) and α-tocopherol (vitamin A), are shown in Figure 2. Both lipids belong to the class of prenol lipids, which is a group of lipids that has been mentioned as molecules of interest in the search for extraterrestrial life by the Europa Lander Science Team (https://europa.nasa.gov/resources/58/europa-lander-study-2016-report/, accessed on 11 May 2022). The parent ion ($m/z$ 430) was observed for α-tocopherol, while the main contributing peak for phylloquinone was $m/z$ 448 ([M-2]$^+$). Multiple (small) fragments were also observed for both lipids, as well as an adduct peak at [M+16]$^+$, most probably caused by oxidation in the laser desorption plume by oxygen atoms available in the sample material or on the holder. Each lipid comes with its own unique mass spectral fingerprint, which can subsequently be used for identification in complex samples. Both lipids were measured with high sensitivity (LOD$_{3\sigma}$ below 100 fmol mm$^{-2}$), as was expected with their aromatic features.

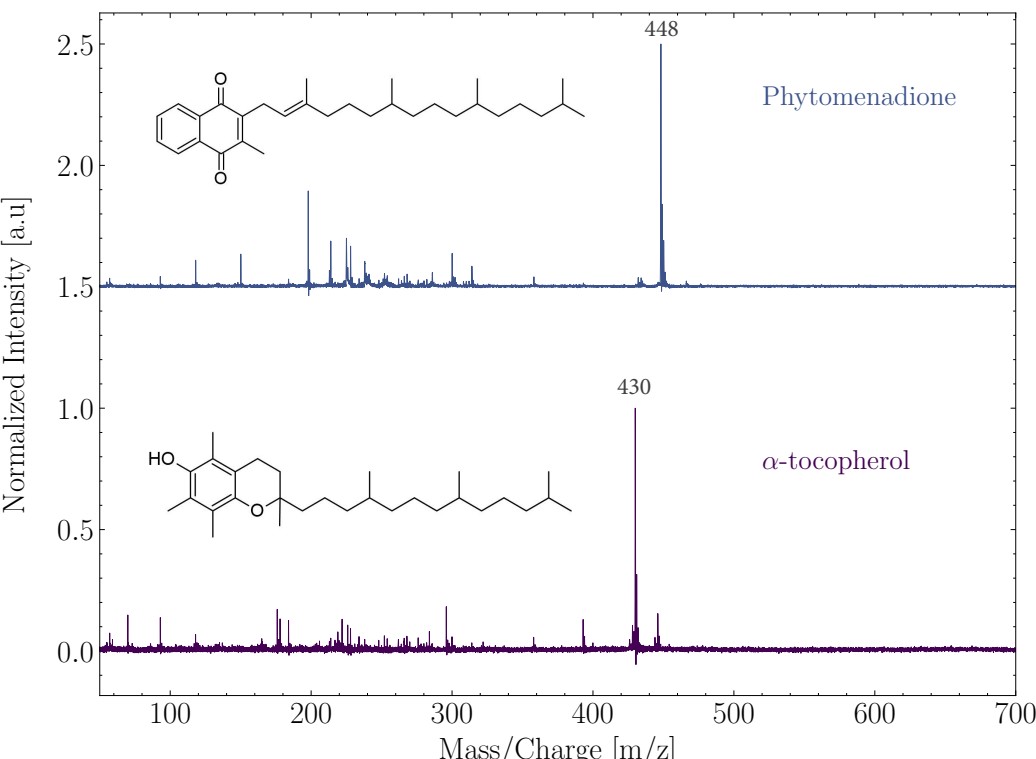

**Figure 2.** Normalized spectra of the lipids phytomenadione and *α*-tocopherol. The structure of the respective lipids was added to the left and prominent mass peaks are indicated. The spectra were offset from each other.

While the above examples demonstrate ORIGIN's ability to detect and identify biomolecules, these molecules are inevitably derived from an Earth-based biochemistry and may not be representative of Venusian biochemistry in sulphuric acid droplets. Performance analysis of ORIGIN with molecules that can survive concentrated sulphuric acid and are presumed to be of biochemical relevance in Venus's cloud deck is desired. A number of such studies are being conducted, and they may provide useful analog material. Studies of Venusian atmospheric prebiotic chemistry in Miller–Urey-type experiments are underway (Cleaves et al. in preparation, private communication, see also [23], their Appendix A.4); such studies are essential to be able to distinguish between biotic and abiotic chemistry. At the same time, we want to understand which kinds of biomolecules may survive in concentrated sulphuric acid.

In this context, cell membrane formation is essential for life as we know it. Duzdevich et al. (in preparation, private communication, see also [23], their Appendix A.3) demonstrated that it is possible to form lipid vesicles in 70% sulphuric acid with simple, saturated-tail lipids, making these species in principle suitable targets for ORIGIN performance analysis. However, these molecules are akin to fatty acids and consist primarily of long unsaturated carbon chains. This is problematic for ORIGIN as it has low detection sensitivity to fatty acids with its current 266 nm laser desorption source. A switch to longer laser wavelengths in the green (e.g., 532 nm) or infrared (e.g., 1064 nm) region might show better performance for these species if these molecules are better able to absorb photons at these wavelengths. These changes are scheduled for future studies. Given that 1064 nm is the fundamental wavelength of the laser, and 532 nm and 266 nm are harmonics, such a switch is easily accomplished. On a space mission, it is possible to implement two miniature microchip lasers at different wavelengths to facilitate such studies.

There are many more potentially bio-relevant molecular classes that can be studied by ORIGIN, but a particularly attractive one is photosynthetic pigments. A long-standing mystery is the unknown UV absorber in Venus's atmosphere [12]. A biological explana-

tion for this absorption can be pigments such as chlorophyll a and phycocyanin, which readily absorb UV radiation and protect organisms from damaging UV radiation or drive photosynthesis [13]. Pigments are therefore an excellent target for VLF science and future performance studies with ORIGIN. Because many pigments contain aromatic ring structures, to which ORIGIN has shown particular sensitivity, it is highly likely that ORIGIN will be able to detect pigments.

### 3.2. Polycyclic Aromatic Hydrocarbons

In the clouds of Venus, large polycyclic aromatic aggregate particles can be formed in concentrated sulphuric acid from small organic molecules via carbon polymerization [27]. Part of this reaction results in the formation of larger polycyclic aromatic compounds, which, after the evaporation of the concentrated sulphuric acid from aerosols, can result in porous particles falling from the clouds. These carbon particles then contribute to the lower haze and are further decomposed, resulting in CO formation [27].

Polycyclic aromatic hydrocarbons (PAHs) are not directly synthesized by biological processes [28], but could nevertheless be important in the search for extraterrestrial life, as they could be precursors of molecules important for the emergence of life [29]. Derivatives of PAHs were found to have a stabilizing influence on fatty acid membranes [30].

Certain PAHs were postulated in 1965 by Pollack and Sagan [31] to be stable in the Venusian atmosphere. Furthermore, Handa [32] and Bains et al. [33] showed that several PAHs and their quinones are able to form stable solutions in concentrated sulphuric acid, which is present as a liquid in the clouds of Venus.

ORIGIN has proven capable of detecting representative PAHs at different concentrations and laser pulse energies, showing its capabilities in detecting and identifying aromatic structures [34]. Anthracene ($C_{14}H_{10}$), pyrene ($C_{16}H_{10}$), perylene ($C_{20}H_{12}$), and corononene ($C_{24}H_{12}$) were used in this study, and examples of these measurements are shown in Figure 3. ORIGIN is sensitive to these PAHs and readily achieves LODs of $\sim$10 femtomol mm$^{-2}$ for these molecules. PAHs show no to limited fragmentation, but have been observed to protonate or cluster. If PAHs are present in sulphuric acid droplets in Venus's clouds, ORIGIN is able to detect them and provide valuable information on the Venusian atmospheric carbon cycle and the habitability of its clouds.

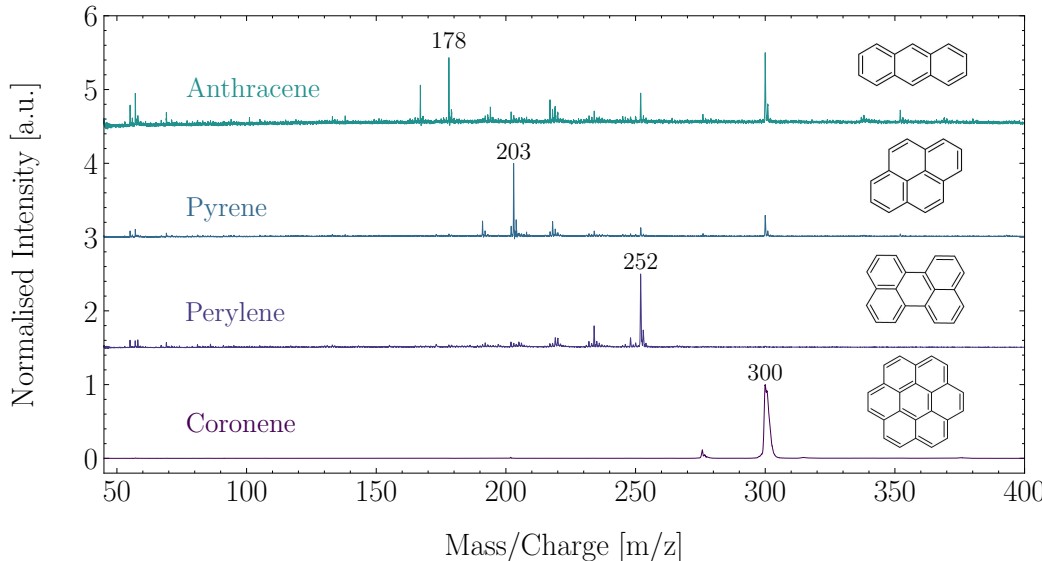

**Figure 3.** Measurement of four representative PAHs with ORIGIN, measured at a laser pulse energy of 4 μJ.

### 3.3. Salts

Venus's atmosphere presents many opportunities to produce salts that can in turn affect the atmospheric chemistry. Hydroxide salts can moderate the pH of sulphuric acid droplets and aid the depletion of gaseous $SO_2$ [35]. Organisms in sulphuric acid droplets could use water and $N_2$ to produce ammonia and oxygen, and then form ammonium sulphate (($NH_4)_2SO_4$) to moderate the pH [16]. Simultaneously, this mechanism explains the relatively high observed oxygen abundances in Venus's atmosphere [36,37]. Finally, the $CO_2$-rich atmosphere presents opportunities to form carbonates ($CO_3^{2-}$). The large, non-spherical Mode 3 particles present in the atmosphere of Venus have been suggested to be crystalline particles consisting of hydrated salts [38,39].

ORIGIN can detect salts as well, as was shown for NaCl and KCl, salts that are likely present on Europa's surface, in Ligterink et al. [26]. In measurements presented in this work, we test whether ORIGIN can detect the salts ($NH_4)_2SO_4$, magnesium sulphate ($MgSO_4$), and calcium carbonate ($CaCO_3$). The ability to detect these and related salts with ORIGIN will help in assessing the habitability of sulphuric acid droplets, further elucidate the chemistry of the Venusian atmosphere and its droplets, and may even be used as an indicator for the presence of life.

The salts were all purchased from Sigma-Aldrich and they have a purity of $\geq$99%. With MilliQ water, 1 M solutions of ($NH_4)_2SO_4$ and $CaCO_3$ were made, while a $MgSO_4$ solution of 2.5 M was used. One μL of the $CaCO_3$ and $MgSO_4$ solutions was deposited in a cavity of a stainless steel sample holder, whereas 4 μL of the ($NH_4)_2SO_4$ solution. Due to the high surface tension of these solutions, the droplets did not fill the cavity, and consequently the salts were not distributed over the cavity after the water evaporated, instead forming concentrated spots in the centre of the cavity. The sample holder was introduced into the ORIGIN setup and a 40 laser position grid was scanned with 6 μJ laser pulses at a wavelength of 266 nm.

The results of these scans are shown in Figure 4. The cation of each salt, $NH_4^+$, $Mg^+$, and $Ca^+$, is detected. Masses belonging to $SO_4$ and $CO_3$ are not detected, which presumably is because these fragments are present as neutral or negatively charged particles. Anions were not detected by ORIGIN as it was operating in positive ion mode. There are no mass peaks that correspond to the ionized intact salt or an ionized salt cluster. This contrasts with the case of NaCl, where in ORIGIN measurements the $Na_2Cl^+$ ion was identified besides the $Na^+$ ion [26]. It is important to note that a relatively high pulse energy of 6 μJ is used for these measurements. A quasi-ablation regime is entered at this moderately high pulse energy.

The ability to detect these salt cations will significantly aid investigations of the Venusian atmosphere. The detection of magnesium and calcium, and other atomic ions, provides insight into the metals available to an organism in sulphuric acid droplets. The identification of $NH_4^+$ can hint at an efficient production mechanism of $NH_3$ and a way to moderate sulphuric acid droplet pH, which can make these environments habitable. Detecting high concentrations of salts in Mode 3 particles can help elucidate their nature. The analysis of salts will be further studied in future work.

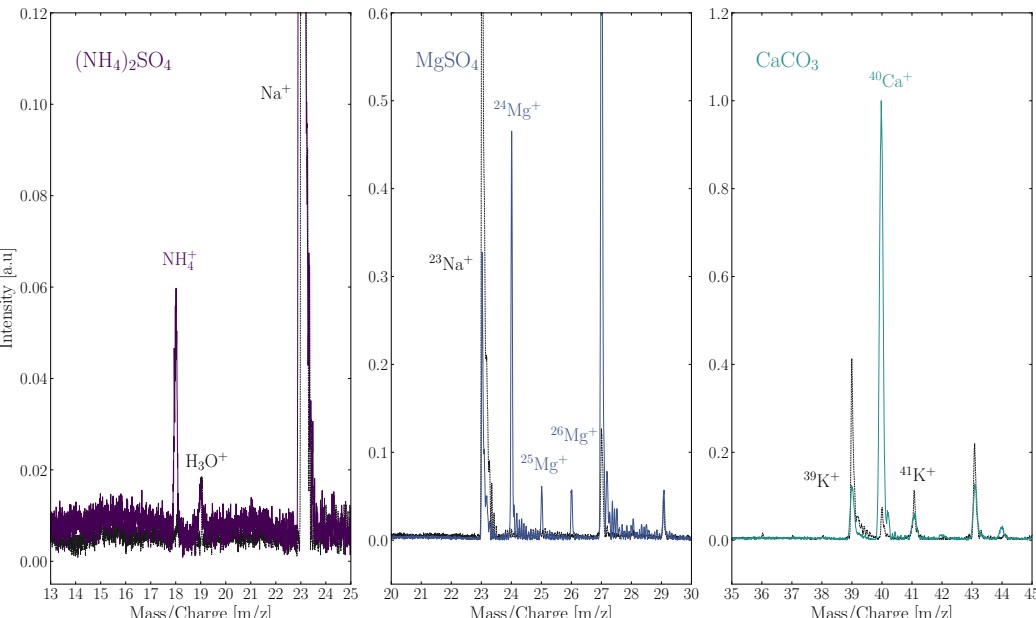

**Figure 4.** Three salts measured by ORIGIN: the $NH_4^+$ ion from the $(NH_4)_2SO_4$ salt (**left**); the $Mg^+$ ion and its isotopes of the $MgSO_4$ salt (**middle**); the $Ca^+$ ion of the $CaCO_3$ salt (**right**). The ionized intact salts, the sulphate and carbonate groups, or any clusters containing these fragments were not detected. A grey dotted line of a blank measurement is added in each panel.

### 3.4. Metals

Large biomolecules, such as proteins, often incorporate metal atoms or ions into their structures to enhance their functionality. Specifically, to catalyse biochemical reactions, metals are essential. Since metals are not easily accessible to any aerial biosphere, it is important to determine whether metals are sufficiently abundant in Venus's atmosphere for life to thrive [6]. One source of metals is meteorites. These objects enter the atmosphere and burn up, thus adding metals locally.

Simultaneously, element analysis can provide insight into other open questions, such as the nature of the unknown UV absorber. There are several candidate molecules that can cause this absorption, such as $S_2O$, $S_2O_2$, or $FeCl_3$, and it may have even more than one contributor [40]. The analysis of the atomic composition can shed light on the identity of the absorber. Specific ratios of atoms (e.g., a 1:3 ratio of Fe to Cl) or the absence of an atom (e.g., low abundances of Fe) can hint at the presence of a specific absorber or rule it out.

By increasing its laser pulse energy and entering the laser ablation regime, ORIGIN can be used to determine metal abundances in the residues of sulphuric acid droplets. When operating in LIMS mode, this instrument has successfully been used to analyse the atomic composition of metal and mineral standards (e.g., [41–43]), and metal ablation is routinely used in ORIGIN's mass calibration procedure. Element analysis of sulphuric acid droplet residues is therefore feasible, but care needs to be taken in the sample handling process. During the ablation step, not only the residue is ablated, but also the underlying sample holder substrate, which will add mass lines to the recorded signal. Sample holders with a well-defined atomic composition are required and usually an elementally pure material is preferred, for example gold.

### 3.5. Sulphur Isotope Fractionation

In an S-rich environment like the sulphuric acid droplets in Venus's clouds, it is not unlikely to find organisms that metabolize sulphur. Sulphate-reducing bacteria can convert $SO_4^{2-}$ to sulphides ($S^{2-}$) [44]. In this process, the light sulphur isotope is favoured, resulting in the depletion of $^{32}S$ in sulphates and the enrichment of $^{32}S$ in the sulphides, while thermochemical processes on Earth can cause sulphur isotope fractionation at levels of

$\delta^{34}$S = ∼20‰, bacteria can fractionate sulphur down to $\delta^{34}$S = −70‰ [45,46]. Therefore, isotope ratios of sulphur can be used as an indicator for biological activity.

In laser ablation mode, the instrument has been used for element and isotope analysis of sulphur in a variety of representative samples and demonstrated its ability to detect isotope fractionation at a level of per mille in these samples [47]. The achieved accuracy of the isotope measurements was sufficient to distinguish between abiotic and biotic fractionation processes. While a femtosecond laser was used for ablation in these experiments to atomize sample material, it was demonstrated earlier that, with a high-power nanosecond laser pulse isotope, abundances can be measured with high accuracy [43].

### 3.6. Single Microbe Detection

Laser ablation can also be used for an elemental analysis of matter, including microbes [25,48]. In a study by Stevens et al. [49], a Martian mudstone analog was inoculated with the *Bacillus subtilis* microbe and analysed with LIMS. Various analysed positions showed telltale element enhancements with respect to the composition of the mineral matrix, indicative of the presence of microbes [50]. This technique allowed for the detection of single microbes with very high detection efficiencies, thus allowing for the investigation of sparse life, as can be expected for the situation in the Venus cloud deck.

ORIGIN, operating in laser ablation mode, can do the same on a VLF mission. By analysing the element compositions of individual droplets in a step-wise fashion, those hosting microorganisms can be distinguished from sterile droplets due to enhanced levels of biologically relevant elements; this kind of analysis will require a specific sample introduction mechanism, which is presented in Section 5.

## 4. Advanced Analysis Techniques

The spectra obtained with the ORIGIN space instrument are analysed to determine the chemical composition of a sample. Traditionally, a fitting routine or variation thereof is employed, where reference mass spectra of individual molecules are matched and fitted to a measured spectrum to determine which molecules are present. While functional, this is a labour-intensive process that requires many individual molecules to be measured to create a database. The grid scanning technique used by the ORIGIN space instrument presents alternative or complementary ways to analyse the data. The grid scan yields a substantial amount of data for which novel statistical analysis techniques can be used to yield more insight into studied samples.

One such example is correlation network analysis [51]. The basis of this technique relies on the fact that surface concentrations of molecules vary from position to position in a sample, but their fragmentation patterns do not. This means that mass fragments belonging to molecule A can be correlated by plotting their signal intensities from different positions against each other. The same can be done for molecule B, but since there will be variations in the surface concentrations between molecule A and B, there will be no correlation between the molecular fragments of A and B. The correlations can be determined for all masses present in a spectrum and visualized in a network. Such a network, therefore, summarizes which masses strongly or weakly correlate with each other in a very comprehensible manner. The strongly correlated masses are likely have the same origin, presumably a common parent molecule. Schwander et al. [51] recently demonstrated on test data that molecules can in fact be distinguished in this way and were also able to visualize organic and refractory contributions in the mass spectrum of a semi-natural permafrost measurement in a correlation network. Correlation network analysis can simplify the analysis of complex mixtures and help narrow down the origin of mass signals by identifying the other masses they correlate with.

When measuring samples that are not prepared in the laboratory, it is, however, plausible that not every compound will be detected in every spectrum because of the sample's inhomogeneity. The fragmentation masses of a certain compound will not correlate in spectra in which the corresponding compound was not detected and will weaken the overall

correlation between these masses the more often a non-detection occurs. It is therefore advisable to first apply a filtering mechanism to separate spectra of different chemical compositions. A machine learning routine based on data clustering techniques has successfully been applied for detailed data analysis [52] to separate the different minerals and fossils present in the Gunflint sample, a 1.88 Ga geological sample from Canada. These measurements were made with the space-prototype instrument operating in LIMS mode. The sample hosts remnants of fossilized microbes that have a distinct composition from the surrounding sample material and which were also well separated from the sample host by the machine learning model. Simply put, this model determines the similarity between spectra and groups spectra that share a certain similarity together in a cluster. Through this, it is possible to separate spectra with distinct chemical compositions and conduct further analysis on the distinct groups. Furthermore, here, a network can help to visualize the similarity relationships between the analysed spectra. By training such a machine learning model on a mineralogical data set, it is therefore possible to later determine the mineralogical composition of unknown geological samples and look for remnants of life. While the above studies focused on minerals, the exact same routines can also be employed on residues of Venusian cloud droplets that are prepared for ORIGIN analysis.

For geological samples, one has the advantage that even when the sample is inhomogeneous in the sense of being composed of a mixture of minerals, the minerals themselves most likely still form crystals which are big enough to yield chemically distinct spectra for different sampling locations. Hence, the mixing will not be very pronounced, and the machine learning model described above will be able to separate the spectra obtained from the different minerals. However, when measuring residues that contain a mixture of certain molecules, each spectrum will contain mass peaks from various components, with the possibility of overlaying fragmentation patterns. The machine learning routine described above cannot deal well with mixtures, and its application is therefore limited in this case. Correlation network analysis, on the other hand, cannot handle overlaying fragmentation patterns very well. Therefore, in this case, one can make use of neural networks, which can be trained to recognize specific fragmentation patterns. By feeding it spectral data both from single compounds and mixtures, the neural network can be trained to handle both cases. Furthermore, it can be trained on both simulated data and data obtained with the ORIGIN instrument in the laboratory to make it more flexible. Good feature selection is crucial. Neural network analysis of ORIGIN data is currently ongoing.

## 5. ORIGIN Flight Design and VLF Implementation

ORIGIN is part of a larger family of similar instruments that share components and technologies. In recent years, various systems have been designed and constructed based on ORIGIN-like instruments. The LAZMA instrument [53], a laser ablation/ionization mass spectrometer (LIMS), flew on the Phobos-Grunt mission (Roscosmos), where it was supposed to study the chemical composition of Phobos regolith. This mission was unfortunately terminated in the Earth's atmosphere. For the proposed MarcoPolo-R mission (ESA), to land on an asteroid, the CAMAM instrument was designed [54]. This LIMS instrument integrated a microscope-camera system for optical analysis of the samples with the laser mass spectrometer. Its Sample Collecting and Introduction System (SCIS) consisted of a polarized metal conveyor band that could collect levitating asteroidal dust particles and transport them to the instrument. To visualize what a flight version of the ORIGIN instrument for a VLF might look like, a CAD drawing is presented in Figure 5. This drawing depicts a concept design for an ORIGIN-like LIMS instrument for in situ studies of the lunar surface.

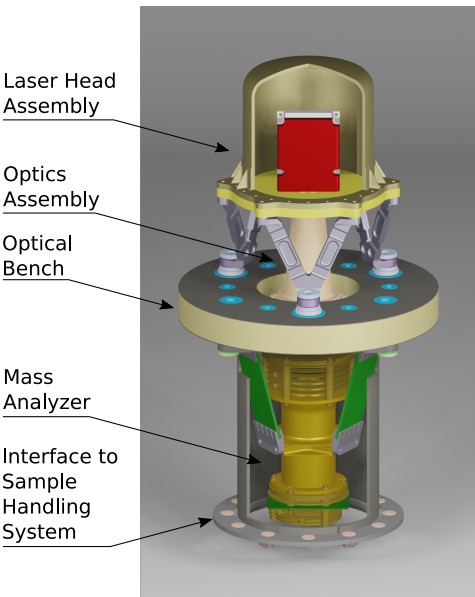

**Figure 5.** CAD drawing of a concept design of an LIMS space instrument for studies of the lunar surface. A design similar to this can also be used for an ORIGIN instrument on the VLF mission.

For the integration of ORIGIN into a VLF mission, design experiences and heritage from these previous missions and designs can be leveraged. The core components of the ORIGIN flight instrument will be a microchip laser and optics, a miniature mass analyser, acquisition electronics and power supplies, a sample preparation and handling system, and a vacuum pumping system. The system can be expanded with, e.g., additional lasers at different wavelengths or an integrated microscope-camera system, at the trade-off of higher instrument weight, volume, and power consumption. An overview of these components and their important parameters is given in Table 1. The weight and power consumption of an ORIGIN-VLF base and expanded instrument stay well within the constraints set for an LDMS package by the VLF mission study [23], even when a sample handling system and 20% margin are included.

**Table 1.** Components of the ORIGIN instrument for flight on VLF.

| Component | Weight (g) | Power Consumption (W) | Dimension (mm) (h × w × d) |
|---|---|---|---|
| Microchip laser | 150 | 2 | 25 × 25 × 25 |
| Laser optics | 120 | 0 | 50 × 35 × 35 |
| Electronics (data acquisition, DPU) | 1400 | 11.4 | 180 × 240 × 60 |
| Mass analyser | 500 | 1.9 | 120 × 60 × 60 |
| Vacuum system and pump | 1500 | 9 | 150 × 70 × 70 |
| Microscope-camera | 190 | 2 | 115 × 65 × 55 |
| Sample collection and introduction system | 2000 | 8 ** | – |
| Support structure and housing | 4000 | – | – |
| Base system * | 11,600 | 36 | – |
| Expanded system * | 12,000 | 50 | – |

The base system consists of only the essential components (single microchip laser, acquisition card, interfaces, mass analyser, pumping, and sample introduction system), whereas the expanded system also incorporates a microscope-camera system and a second microchip laser at a different wavelength. * A 20% margin on the weight and power consumption is included in these systems. ** Power only needed during sample collection.

The exact ORIGIN-VLF flight instrument design must be tailored to the mission parameters, some of which are detailed in the VLF mission study [23]. To protect from the Venusian atmosphere, specifically the sulphuric acid droplets, all science instruments will

be housed in a pressure vessel. An ORIGIN-VLF flight instrument therefore does not have to use shielding of its own to protect its components. Since the ORIGIN mass analyser can only operate in a vacuum of less than $1 \times 10^{-6}$ mbar, a light-weight vacuum chamber with a pumping system needs to be added. This requires an ultra-high-vacuum pump. Miniature space-adopted turbomolecular pumps are available and will suffice [55].

Sample material, in the form of cloud droplets, will be collected by the Venus Cyclone Sampler (VCS) or Venus Fog Harp (VFH), concentrated, and delivered to the instrument (ref. [23], their section C.4.2). A critical component of an ORIGIN flight instrument is a sample preparation and handling system, since sample material needs to be transferred from the pressure vessel to the ORIGIN vacuum chamber. As described in the VLF mission study, the Solid Sample Delivery System (SSDS) can be used. Just like SCIS on the CAMAM instrument, this delivery system relies on a conveyor-belt-like system. Sample material is placed on a tape and left to dry. The sample material is moved into the analysis region of ORIGIN, and by moving the tape up with an elevator, it is pressed into a knife edge seal to make a vacuum seal, and subsequently a high vacuum can be generated.

Alternatively, an SCIS-like mechanism can be used with a differentially pumped feedthrough for the conveyor belt. Differential pumping makes use of different stages of vacuum between which there is a pressure gradient. This is particularly useful when material needs to be transported from, e.g., an ambient pressure environment to a high-vacuum environment without breaking the vacuum. Direct transfer through a small opening in a high-vacuum chamber is possible, but requires a large pumping capacity to maintain the vacuum. Instead, transfer through an intermediate low-vacuum chamber ensures that, on the whole, a smaller amount of pumping capacity is required. This concept can be employed for the ORIGIN sample handling system. Samples are placed on the conveyor belt in the atmosphere of the pressure vessel, pass through a small opening to the differentially pumped chamber that is maintained at (sub-) mbar pressure, and finally pass through another small opening to the main instrument chamber at high-vacuum pressure. The openings at each transfer point are small and only have to be slightly larger than the width of the conveyor belt and the combined height of the belt and sample material. At each transfer point, there is a shutter that can be closed. The pumping system and the instrument can operate without closing these shutters, but closing them can be beneficial to achieve a slightly better vacuum when desired or to maintain a vacuum in the chamber while the pumps are shut off (e.g., during cruise phase).

This differentially pumped system has three benefits. A continuous vacuum ensures that the interior of the ORIGIN flight instrument remains clean and its components (e.g., the miniature mass analyser, microchip laser) are never exposed to any vapours that might be present within the pressure vessel. At the same time, the still liquid sulphuric acid droplet can be placed under vacuum in the feedthrough chamber. Sulphuric acid in concentrated form is difficult to evaporate due to its low vapour pressure (e.g., [56,57]). However, under vacuum, its boiling point is lowered, and this will speed up the $H_2SO_4$ evaporation. Finally, and most importantly, this system gives more flexibility to move sample material into the laser focus. The ORIGIN methodology relies to a large extent on its grid scan methods, whereby the sample holder moves in the XY plane to place material in the laser focus. Samples may segregate when liquids dry, and this method is essential to analyse the different sample components. At the same time, grid scan data can be stacked to obtain an average overview of the chemical composition of the sample and increase the sensitivity. A differentially pumped feedthrough makes it possible to perform a grid scan, as sample material can be moved in the X-plane by moving the conveyor-belt back and forth and in the Y-plane by moving the entire conveyor system sideways. A schematic depiction of ORIGIN with a differentially pumped sample introduction system is presented in Figure 6.

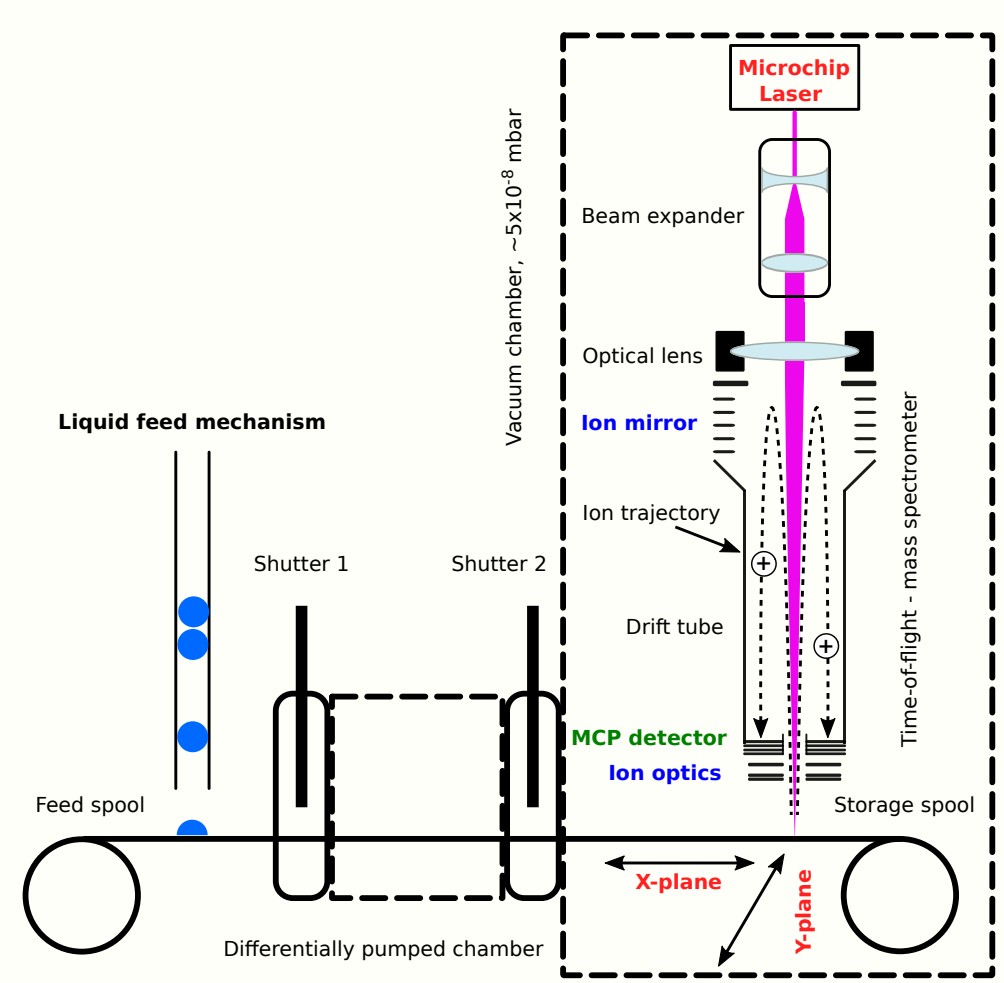

**Figure 6.** Schematic depiction of an ORIGIN-VLF flight instrument with a differentially pumped sample feedthrough mechanism. The mass analyser and microchip laser are housed in a vacuum chamber. A conveyor-belt sample introduction system transfers sample material from outside the vacuum chamber through a differentially pumped feedthrough mechanism to near the laser focal point.

The differentially pumped feedthrough mechanism provides another exciting possibility. Instead of internally feeding it liquid droplets from the VCS or VFH, the instrument can be integrated into the exterior shell of the pressure vessel and the conveyor belt tape can be exposed directly to the Venusian atmosphere. Here, sulphuric acid droplets will accumulate on the tape. By adopting conservative values used for the VFH [23], such as 3% collection efficiency, a wind flow rate of 0.25 m s$^{-1}$, VFH collection area of 400 cm$^{-2}$, and large droplet density of 10 cm$^{-3}$, which result in a VFH droplet collection rate of $3 \times 10^6$ particles min$^{-1}$, we can estimate a collection efficiency for micrometer-sized particles of about 7500 particles min$^{-1}$ cm$^{-2}$. An Individual Droplet Collection Mechanism (IDCM)—a continuously running conveyor belt that is exposed to Venus's atmosphere—can collect many hundreds of thousands of individual droplets over the course of an operation. The VCS or VFH collect droplets and concentrate them into a liquid, diluting habitability and biosignatures and even bringing chemicals together to react and destroy such signatures. An IDCM presents the possibility to analyse individual, isolated, and undiluted sulphuric acid droplets and their chemical content, which may include living organisms. This system thus presents a more robust way to identify molecular and elemental biosignatures, with the trade-off that concentrations are lower and many individual droplets will not be interesting from a search-for-life perspective. For ORIGIN analysis, the selection

of the most scientifically interesting droplets with, for example, a microscope system or fluorescence measurements is recommended.

Control of the ORIGIN instrument during the mission can be fully autonomous. A number of key operations, such as the collection and preparation of sample material or performing a grid scan, consist of simple automated routines that can be handled by the control software. Target selection for the grid scan is more critical, but can also be fully automated. For example, by giving the microscope criteria to select samples on (e.g., residue size, colour, or specific fluorescence signals) and subsequently performing a single position ORIGIN measurement to see if pre-specified or high mass ions (e.g., signals above 100 $m/z$) are present in the sample, the most chemically complex, and presumably most scientifically relevant, samples can be selected for grid scanning. The spectra generated from the grid scan can be treated by the onboard data processing unit to reduce the amount of data (e.g., histogramming) or even analyse it, for example with the techniques described in Section 4. Human intervention might still be desired, however. Target selection depends on predefined criteria, but this may mean that some critical signatures are missed that are easily picked up by a human looking at the data. Data processing of the data with the VLF data processing unit is limited by the algorithms that it is programmed with, and therefore delivery of the raw data to Earth is useful in performing different analysis routines. In the end, the approach taken will depend on parameters such as available data volume, data transmission, and onboard data processing power, which will make either automation or human operation more desirable. From a technical standpoint, ORIGIN can operate in both control regimes.

## 6. Conclusions

ORIGIN is a highly sophisticated laser desorption/laser ablation mass spectrometer that can be deployed on a Venus Life Finder mission to investigate biosignatures and habitability indicators in the atmosphere of Venus. This instrument is best deployed for analysing non-volatile molecules, which may be present in the sulphuric acid droplets that are found in Venus's clouds. ORIGIN has a proven track record in identifying and analysing biomolecules (e.g., amino acids and lipids), polycyclic aromatic hydrocarbons (PAHs), and salts; it can also bring to the fore atomic compositions, isotopes, and even the elemental signatures of microbes. New measurements of ammonium sulphate are highlighted, demonstrating that ORIGIN can detect ammonium ions, which can probably aid in identifying biological pathways for ammonia formation, thereby moderating the pH of sulphuric acid droplets. To further assess ORIGIN's capabilities, trials are suggested that target molecules found in Venus analog chemistry experiments. The implementation of ORIGIN in the Venus Life Finder mission is discussed, focusing in particular on a conveyor-belt-like sample delivery system. This system can be used to deliver residues extracted from accumulated and concentrated sulphuric acid droplets. An alternative is to let this system collect individual droplets and present each of them to ORIGIN for analysis.

**Author Contributions:** Conceptualization, N.F.W.L. and A.R.; methodology, N.F.W.L., A.R., S.G., N.J.B., C.P.d.K. and M.T.; software, N.F.W.L. and P.K.S.; formal analysis, N.F.W.L.; investigation, N.F.W.L.; resources, A.R., P.W. and N.F.W.L.; writing—original draft preparation, N.F.W.L.; writing—review and editing, all authors; visualization, K.A.K., N.F.W.L. and P.K.S.; project administration, A.R.; funding acquisition, A.R., N.F.W.L. and P.W. All authors have read and agreed to the published version of the manuscript.

**Funding:** This research was funded by the Swiss National Science Foundation, including Ambizione grant 193453, and NCCR PlanetS.

**Data Availability Statement:** The data presented in this study are available on request from the corresponding authors. The data are not publicly available due to the large size of the dataset.

**Acknowledgments:** The authors thank D. Duzdevich, H.J. Cleaves, J.J. Petkowski, and S. Seager for helpful discussions and the anonymous reviewer for the useful comments.

**Conflicts of Interest:** The authors declare no conflict of interest.

## Abbreviations

The following abbreviations are used in this manuscript:

| | |
|---|---|
| ORIGIN | ORganics Information Gathering INstrument |
| ESA | European Space Agency |
| JAXA | Japan Aerospace Exploration Agency |
| DAVINCI+ | Deep Atmosphere Venus Investigation of Noble gases, Chemistry, and Imaging |
| VERITAS | Venus Emissivity, Radio Science, InSAR, Topography, and Spectroscopy |
| NASA | National Aeronautics and Space Administration |
| VLF | Venus Life Finder |
| MIT | Massachusetts Institute of Technology |
| GC-MS | Gas chromatography mass spectrometry |
| PAH | Polycyclic aromatic hydrocarbons |
| VAIHL | Venus Airborne Investigation of Habitability and Life |
| LMS | Laser-based mass spectrometer |
| LDMS | Laser desorption/ionization mass spectrometry |
| LIMS | Laser ablation/ionization mass spectrometry |
| CAMAM | Camera and mass spectrometer |
| SCIS | Sample Collecting and Introduction System |
| VCS | Venus Cyclone Sampler |
| VFH | Venus Fog Harp |
| SSDS | Solid Sample Delivery System |
| IDCM | Individual Droplet Collection Mechanism |

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
