# Peer review of "The ORIGIN Space Instrument for Detecting Biosignatures and Habitability Indicators on a Venus Life Finder Mission"

_aerospace, doi:10.3390/aerospace9060312_

Round 1

Reviewer 1 Report

This is a good overview description of the ORIGIN instrument.  The capabilities described are clearly relevant to assessing the possibility of biological activity in the clouds of Venus.  The authors have a proper understanding of the environment of the target atmosphere (lower cloud layer) and the challenges it presents for in situ sample collection and analysis. ORIGIN is obviously a versatile instrument, capable of focusing on different properties and adjusting to altered priorities during the course of a mission as envisioned for VAIHL, which is a well-designed (though not the only plausible) mission appropriate for assessing the astrobiological potential of the Venusian atmosphere.

This paper has no major flaws that I recognize, but I would draw the authors’ attention to the following, in more-or-less descending order of concern:

  1. In view of the large number of acronyms used, a listing with their definitions would be helpful.
  2. A number of different operations, each requiring a decision for their implementation and control, are available on ORIGIN. These include activation and control of grid scanning, adjusting the plane of focus, changing the laser wavelength, switching back and forth between the laser ablation mode, and “selection of the most scientifically interesting droplets.” There is no explanation of how these decisions will be made in real time and how their control will be implemented (e.g., by remote control from Earth-based operators, or by automated on-board algorithms?).  These details may lie outside the scope of the present paper, but if they ultimately could have a bearing on the hardware of the instrument or the software that controls it, they should be mentioned.
  3. The VAIHL mission architecture is designed for 30 days. Even assuming that active sample collection occurs only for a fraction of that time, the total number of samples collected for analysis (which could entail 7500 samples/min/sq cm, and as many as 4000 spectra/sample) could generate an astronomical amount of data.  This is dealt with briefly (by data compression, etc.), but the reader is left wondering whether this could be a concern ― does the engineering of the instrument exceed any ability to analyze the data it yields?  This is not an argument against the instrument, just a point that might warrant brief attention.
  4. Lines 475-6: I don’t see how the numbers given yield the stated result. Clearly I could be missing some relevant factor or be doing the calculation wrong, but I would merely want the authors to double-check their calculations.
  5. Line 307-8: Why is FeCl3 the most likely absorber, as opposed to say S8 (Schulze-Makuch et al. 2004. Astrobiology 4: 11)? Explain or give reference.
  6. Lines 444-5: References should be provided for these collection mechanisms
  7. Line 456: How can a vacuum be maintained with the shutters open?
  8. The paper is well written and for the most part clear. As always, a few minor grammatical and punctuation corrections are needed, including but not limited to the following:
  1. Lines 145-6: Syntax is confusing
  2. Line 244: “. . . were postulated. . .” instead of “. . . have already been postulated. . .”
  3. Lines 247-250: The purpose of this last sentence in the paragraph is not evident.
  4. Line 422: “To visualize what a flight version. . . “
  5. Line 364: Unclear; syntax appears to be faulty
  6. Line 382: “. . . present in the Gunflint sample . . .” and “These data were . . .”

Author Response

We thank the reviewer for their comments. We have addressed the comments of the reviewer below. The original question is indicated with an “<<<” marker, while our reply is indicated with “>>>”. Textual changes in the manuscript have been highlighted in bold.

<<< This is a good overview description of the ORIGIN instrument.  The capabilities described are clearly relevant to assessing the possibility of biological activity in the clouds of Venus.  The authors have a proper understanding of the environment of the target atmosphere (lower cloud layer) and the challenges it presents for in situ sample collection and analysis. ORIGIN is obviously a versatile instrument, capable of focusing on different properties and adjusting to altered priorities during the course of a mission as envisioned for VAIHL, which is a well-designed (though not the only plausible) mission appropriate for assessing the astrobiological potential of the Venusian atmosphere.

<<< This paper has no major flaws that I recognize, but I would draw the authors’ attention to the following, in more-or-less descending order of concern:

<<< In view of the large number of acronyms used, a listing with their definitions would be helpful.

>>> We have added a list of abbreviations and acronyms to the manuscript.

<<< A number of different operations, each requiring a decision for their implementation and control, are available on ORIGIN. These include activation and control of grid scanning, adjusting the plane of focus, changing the laser wavelength, switching back and forth between the laser ablation mode, and “selection of the most scientifically interesting droplets.” There is no explanation of how these decisions will be made in real time and how their control will be implemented (e.g., by remote control from Earth-based operators, or by automated on-board algorithms?).  These details may lie outside the scope of the present paper, but if they ultimately could have a bearing on the hardware of the instrument or the software that controls it, they should be mentioned.

>>> We agree that this is useful information to add and have therefore added a brief paragraph at the bottom of section 5:

“Control of the ORIGIN instrument during the mission can be fully autonomous. A number of key operations, such as the collection and preparation of sample material or performing a grid scan, consist of simple automated routines that can be handled by the control software. Target selection for the grid scan is more critical, but can also be fully automated. For example, by giving the microscope criteria to select samples on (e.g., residue size, color, or specific fluorescence signals) and subsequently performing a single position ORIGIN measurement to see if pre-specified or high mass ions (e.g., signals above 100 m/z) are present in the sample, the most chemically complex, and presumably most scientifically relevant, samples can be selected for grid scanning. The spectra generated from the grid scan can be treated by the data processing unit on the mission to reduce the amount of data (e.g., histogramming) or even analyze it, for example with the techniques described in Sect. 4. Human intervention might still be desired, however. Target selection depends on predefined criteria, but this may mean that some critical signatures are missed that are easily picked up by a human looking at the data. Data processing of the data with the VLF data processing unit is limited by the algorithms that it is programmed with, and therefore delivery of the raw data to Earth is useful to perform different analysis routines. In the end, the approach taken will depend on parameters such as available data volume and onboard data processing power, which will make either automation or human operation more desirable. From a technical standpoint, ORIGIN can operate in both control regimes.”

<<< The VAIHL mission architecture is designed for 30 days. Even assuming that active sample collection occurs only for a fraction of that time, the total number of samples collected for analysis (which could entail 7500 samples/min/sq cm, and as many as 4000 spectra/sample) could generate an astronomical amount of data.  This is dealt with briefly (by data compression, etc.), but the reader is left wondering whether this could be a concern ― does the engineering of the instrument exceed any ability to analyze the data it yields?  This is not an argument against the instrument, just a point that might warrant brief attention.

>>> This is indeed a concern. In the VLF mission study a maximum data downlink of ~60 Mb every ~5 days is mentioned, whereas ORIGIN in its current (non-optimized) laboratory environment generates about 1 GB of data for every 4000 spectra scan. On-mission data processing and compression will be essential, but also the careful selection of targets to not “waste” data volume on uninteresting targets. Besides the additional information presented above, we have also modified the paragraph on data volume in section 2: “A secondary limitation is the generated data volume, which can be substantial if all individual spectra were to be transmitted and much more than the downlink data volume of the VLF can handle [a maximum downlink of 500 MBits per VLF circumnavigation of Venus is given in 23 ]. It is therefore essential to decrease the data volume and this can be done by online data processing such as histogramming to, for example, reduce 4’000 spectra to a single one. Data compression routines can be employed to further reduce the data volume for transfer back to Earth.”

<<< Lines 475-6: I don’t see how the numbers given yield the stated result. Clearly I could be missing some relevant factor or be doing the calculation wrong, but I would merely want the authors to double-check their calculations.

>>> With the estimated Venus conditions given in the VLF mission study report (3% collection efficiency, 0.25 m/s flow rate, VFH selectivity for >1um-sized droplets, and a VFH collection area of 20x20cm) the authors find that it will collect 3M particles / min. Dividing this number by the area of VFH of 400 cm2, results in the listed 7500 droplets min-1 cm-2. To clarify how we calculate the collection rate, we have modified the sentence to: “By adopting conservative values used for the VFH [23], such as 3% collection efficiency, a wind flow rate of 0.25 m s−1, VFH collection area of 400 cm−2, and large droplet density of 10 cm−3, which result in a VFH droplet collection rate of 3E6 particles min−1, we can estimate a collection efficiency for micrometer-sized particles of about 7500 particles min−1 cm−2.”

<<< Line 307-8: Why is FeCl3 the most likely absorber, as opposed to say S8 (Schulze-Makuch et al. 2004. Astrobiology 4: 11)? Explain or give reference.

>>> The editor also commented on this and we have used some of the editors input to modify this part to the following: “There are several candidate molecules that can cause this absorption, such as S2O, S2O2, or FeCl3, and it may have even more than one contributor [Pérez‐Hoyos+2018]. The analysis of the atomic composition can shed light on the identity of the absorber. Specific ratios of atoms (e.g., a 1:3 ratio of Fe to Cl) or the absence of an atom (e.g., low abundances of Fe) can hint at the presence of a specific absorber or rule it out.”

<<< Lines 444-5: References should be provided for these collection mechanisms

>>> We have added a reference to the VLF Mission study report section C.4.2, where these collection mechanisms are introduced.

<<< Line 456: How can a vacuum be maintained with the shutters open?

>>> The brief explanation is that the openings of the transfer points are small, so the pump can overcome the leak rate of the opening. By using a differentially pumped system, where material is transported from atmosphere to a low-vacuum chamber and finally to a high-vacuum chamber, the load on the pumps are reduced.

We realized that our initial description of this system was lacking and have therefore expanded the relevant paragraphs, see also below:

“Alternatively, a SCIS-like mechanism can be used with a differentially pumped feedthrough for the conveyor belt. Differential pumping makes use of different stages of vacuum between which there is a pressure gradient. This is particularly useful when material needs to be transported from, e.g., an ambient pressure environment to a high-vacuum environment without breaking the vacuum. Direct transfer through a small opening in a high-vacuum chamber is possible, but requires a large pumping capacity to maintain the vacuum. Instead, transfer through an intermediate low-vacuum chamber ensures that, on the whole, a smaller amount of pumping capacity is required. This concept can be employed for the ORIGIN sample handling system. Samples are placed on the conveyor belt in the atmosphere of the pressure vessel, pass through a small opening to the differentially pumped chamber that is maintained at (sub-) mbar pressure, and finally pass through another small opening to the main instrument chamber at high-vacuum pressure. The openings at each transfer point are small and only have to be slightly larger than the width of the conveyor belt and the combined height of the belt and sample material. At each transfer point, there is a shutter that can be closed. The pumping system and the instrument can operate without closing these shutters, but closing them can be beneficial to achieve a slightly better vacuum when desired or to maintain a vacuum in the chamber while the pumps are shut off (e.g., during space flight).

This differentially pumped system has three benefits. A continuous vacuum ensures that the interior of the ORIGIN flight instrument remains clean and its components (e.g., the miniature mass analyzer, microchip laser) are never exposed to any vapors that might be present within the pressure vessel. At the same time, the still liquid sulfuric acid droplet can be placed under vacuum in the feedthrough chamber. Sulfuric acid in concentrated form is difficult to evaporate due to its low vapor pressure [e.g. 56, 57 ]. However, under vacuum, its boiling point is lowered, and this will speed up the H2SO4 evaporation. Finally, and most importantly, this system gives more flexibility to move sample material into the laser focus. The ORIGIN methodology relies to a large extent on its grid scan methods, whereby the sample holder moves in the XY plane to place material in the laser focus. Samples may segregate when liquids dry, and this method is essential to analyse the different sample components. At the same time, grid scan data can be stacked to obtain an average overview of the chemical composition of the sample and increase the sensitivity. A differentially pumped feedthrough makes it possible to perform a grid scan, as sample material can be moved in the X-plane by moving the conveyor-belt back and forth and in the Y-plane by moving the entire conveyor system sideways. A schematic depiction of ORIGIN with a differentially pumped sample introduction system is presented in Fig. 6.”

<<< The paper is well written and for the most part clear. As always, a few minor grammatical and punctuation corrections are needed, including but not limited to the following:

<<< Lines 145-6: Syntax is confusing

>>> This sentence has been modified to: “The data volume can be decreased by online data processing such as histogramming to, for example, reduce 4'000 spectra to a single one. Data compression routines can be employed to further reduce the data volume for transfer back to Earth.”

<<< Line 244: “. . . were postulated. . .” instead of “. . . have already been postulated. . .”

>>> We have made the requested change: “Certain PAHs were postulated in 1965 by Pollack and Sagan [31] to be stable in the Venusian atmosphere.”

<<< Lines 247-250: The purpose of this last sentence in the paragraph is not evident.

>>> We have removed this sentence from the manuscript.

<<< Line 422: “To visualize what a flight version. . . “

>>> We have made the suggested change: “To visualize what a flight version of the ORIGIN instrument for a VLF might look like, …”

<<< Line 364: Unclear; syntax appears to be faulty

>>> This sentence has been modified to: “…, there will be no correlation between the molecular fragments of A and B.”

<<< Line 382: “. . . present in the Gunflint sample . . .” and “These data were . . .”

>>> Changes to the sentences have been made: “… to separate the different minerals and fossils present in the Gunflint sample, a 1.88 Ga geological sample from Canada. These data were obtained with the space-prototype instrument operating in LIMS mode.”

Reviewer 2 Report

Overall the MS is acceptable. Still you need to address some typos and some grammar corrections then the MS should be OK for publication. I would still like to advise you that you should proofread the MS carefully.

Author Response

We thank the reviewer for their comments. The indicated typos and grammar mistakes have been fixed. To several more specific comments/questions, we have given a reply below. The original question is indicated with an “<<<” marker, while our reply is indicated with “>>>”. All textual changes in the manuscript have been highlighted in bold.

<<< To make easier for the for the reader, perhaps you would want provide a table of these AAs in the supplementary notes

>>> We have added a footnote in the manuscript with the requested information.

<<< About the “pressure vessel” and the comment: I am bit confused with these two words... is this a noun (pressure vessel) or an adjective as in pressurized vessel? please check and be sure know what you are saying.

>>> In the VLF mission study report, the compartment that will house the instruments is indicated as the “pressure vessel”. We prefer to not deviate from the terminology used in this document.

<<< don't understand what you mean... shutters open and vacuum maintained?

>>> We realized that our initial description of this system was lacking (the other reviewer also commented on it) and have therefore expanded and clarified this paragraph to better explain the mechanism:

“Alternatively, a SCIS-like mechanism can be used with a differentially pumped feedthrough for the conveyor belt. Differential pumping makes use of different stages of vacuum between which there is a pressure gradient. This is particularly useful when material needs to be transported from, e.g., an ambient pressure environment to a high-vacuum environment without breaking the vacuum. Direct transfer through a small opening in a high-vacuum chamber is possible, but requires a large pumping capacity to maintain the vacuum. Instead, transfer through an intermediate low-vacuum chamber ensures that, on the whole, a smaller amount of pumping capacity is required. This concept can be employed for the ORIGIN sample handling system. Samples are placed on the conveyor belt in the atmosphere of the pressure vessel, pass through a small opening to the differentially pumped chamber that is maintained at (sub-) mbar pressure, and finally pass through another small opening to the main instrument chamber at high-vacuum pressure. The openings at each transfer point are small and only have to be slightly larger than the width of the conveyor belt and the combined height of the belt and sample material. At each transfer point, there is a shutter that can be closed. The pumping system and the instrument can operate without closing these shutters, but closing them can be beneficial to achieve a slightly better vacuum when desired or to maintain a vacuum in the chamber while the pumps are shut off (e.g., during space flight).

This differentially pumped system has three benefits. A continuous vacuum ensures that the interior of the ORIGIN flight instrument remains clean and its components (e.g., the miniature mass analyzer, microchip laser) are never exposed to any vapors that might be present within the pressure vessel. At the same time, the still liquid sulfuric acid droplet can be placed under vacuum in the feedthrough chamber. Sulfuric acid in concentrated form is difficult to evaporate due to its low vapor pressure [e.g. 56, 57 ]. However, under vacuum, its boiling point is lowered, and this will speed up the H2SO4 evaporation. Finally, and most importantly, this system gives more flexibility to move sample material into the laser focus. The ORIGIN methodology relies to a large extent on its grid scan methods, whereby the sample holder moves in the XY plane to place material in the laser focus. Samples may segregate when liquids dry, and this method is essential to analyse the different sample components. At the same time, grid scan data can be stacked to obtain an average overview of the chemical composition of the sample and increase the sensitivity. A differentially pumped feedthrough makes it possible to perform a grid scan, as sample material can be moved in the X-plane by moving the conveyor-belt back and forth and in the Y-plane by moving the entire conveyor system sideways. A schematic depiction of ORIGIN with a differentially pumped sample introduction system is presented in Fig. 6.”